# Longitudinal Analysis of Neutralizing Potency against SARS-CoV-2 in the Recovered Patients after Treatment with or without Favipiravir

**DOI:** 10.3390/v14040670

**Published:** 2022-03-24

**Authors:** Kanako Shinada, Takashi Sato, Saya Moriyama, Yu Adachi, Masahiro Shinoda, Shinichiro Ota, Miwa Morikawa, Masamichi Mineshita, Takayuki Matsumura, Yoshimasa Takahashi, Masaharu Shinkai

**Affiliations:** 1Tokyo Shinagawa Hospital, Tokyo 140-8522, Japan; kanakoshinada7@gmail.com (K.S.); mshinopy@gmail.com (M.S.); shin.ohta0915@gmail.com (S.O.); miwapicco@outlook.jp (M.M.); shinkai050169@gmail.com (M.S.); 2Department of Internal Medicine, Division of Respiratory Medicine, St. Marianna University School of Medicine, Kanagawa 216-8511, Japan; m-mine@marianna-u.ac.jp; 3Research Center for Drug and Vaccine Development, National Institute of Infectious Diseases, Tokyo 162-8640, Japan; sayamrym@niid.go.jp (S.M.); yuadachi@niid.go.jp (Y.A.); matt@niid.go.jp (T.M.); ytakahas@niid.go.jp (Y.T.)

**Keywords:** favipiravir, COVID-19, SARS-CoV-2, neutralizing antibody, neutralizing potency index, neutralization breadth index

## Abstract

The effect of treatment with favipiravir, an antiviral purine nucleoside analog, for coronavirus disease 2019 (COVID-19) on the production and duration of neutralizing antibodies for SARS-CoV-2 was explored. There were 17 age-, gender-, and body mass index-matched pairs of favipiravir treated versus control selected from a total of 99 patients recovered from moderate COVID-19. These subjects participated in the longitudinal (>6 months) analysis of (i) SARS-CoV-2 spike protein’s receptor-binding domain IgG, (ii) virus neutralization assay using authentic virus, and (iii) neutralization potency against original (WT) SARS-CoV-2 and cross-neutralization against B.1.351 (beta) variant carrying triple mutations of K417N, E484K, and N501Y. The results demonstrate that the use of favipiravir: (1) significantly accelerated the elimination of SARS-CoV-2 in the case vs. control groups (*p* = 0.027), (2) preserved the generation and persistence of neutralizing antibodies in the host, and (3) did not interfere the maturation of neutralizing potency of anti-SARS-CoV-2 and neutralizing breadth against SARS-CoV-2 variants. In conclusion, treatment of COVID-19 with favipiravir accelerates viral clearance and does not interfere the generation or maturation of neutralizing potency against both WT SARS-CoV-2 and its variants.

## 1. Introduction

Coronavirus disease 2019 (COVID-19) has infected over 240 million patients worldwide [1]. Efforts to repurpose currently available antiviral drugs or anti-inflammatory/immunomodulatory agents for the treatment of COVID-19 is being widely evaluated [2]. Of these, favipiravir, a selective inhibitor of viral RNA-dependent RNA polymerase, approved for emerging/reemerging or resistant influenza virus infection, has been examined. Its activity against SARS-CoV-2 was predicted based on its ability to neutralize the virus in vitro and several clinical trials demonstrating more rapid viral clearance and shorter febrile periods [3,4]. Based on these reports, several phase 3 clinical trials of randomized, placebo control studies of favipiravir in COVID-19 patients have been performed in the US and Japan [5,6]. As of 17 March 2022, a total of 24 phase 3 clinical studies exploring the effect of favipiravir on COVID-19 in over 20 countries were registered at ClinicalTrials.gov [7]. A recent meta-analysis of 9 favipiravir clinical trials showed significant clinical improvement within 7 days of hospitalization in the favipiravir group (*p* = 0.001 vs. control group) [8]. As for the antiviral effects of favipiravir, faster viral clearance was observed; although, the difference did not reach statistical significance (*p* = 0.094) in this meta-analysis [8]. A more sophisticated phase 3 clinical trial with a larger sample size involving early-onset COVID-19 patients with risk factors has been initiated to examine these issues in greater detail [9,10].

The antiviral effects of favipiravir therapy were evaluated as primary or secondary endpoints including (1) time to resolution of hypoxia, (2) time to alleviation of symptoms, (3) negative conversion of detectable SARS-CoV-2, and (4) changes in patients’ clinical status/chest X-ray findings. However, antiviral therapy could affect the host immune response by decreasing the amount and duration of viral antigen, potentially influencing subsequent susceptibility to reinfection. For example, treatment with anti-influenza virus drugs reduced production of mucosal secretory IgA and protective Abs at both early (21 days) and late (60 days) times after influenza infection in murine models, that may account for the higher reinfection rates observed in patients treated with oseltamivir or zanamivir vs. untreated controls the following year [11,12,13].

While compassionate investigational use of favipiravir would be favored in this emerging/pandemic situation, it is important to determine whether favipiravir affects host responsiveness to subsequent infection. Numerous reports demonstrate that protection in humans and animals by COVID-19 vaccines is mediated by neutralizing antibody [14]. Indeed, the US Food and Drug Administration authorized the use of neutralizing monoclonal Abs against COVID-19 for early therapy of individuals at high risk of severe disease [15,16]. The key to the neutralization of SARS-CoV-2 are Abs specific for the receptor binding domain (RBD) by blocking cell entry of SARS-CoV-2, while the lower levels of anti-RBD Abs associated with mild disease and/or shorter duration of symptoms [17,18]. Therefore, concerns about the magnitude of host immune activity against COVID-19 after favipiravir therapy should be addressed.

A recent report demonstrated that the higher levels of anti-RBD Ab observed in patients with severe COVID-19 did not necessarily correlate with enhanced neutralization [19]. Instead, a newly proposed ‘neutralizing potency index’ (NPI) more accurately predicted protection. The NPI increased with time during the convalescent phase despite an anti-RBD Ab decay [19,20]. Another concern is the emergence of SARS-CoV-2 variants such as B.1.1.7 (alpha), B.1.351 (beta), B.1.617.2 (delta), and highly transmissible B.1.1.529 (omicron) [21,22,23,24]. As these variants could escape from acquired humoral immunity of the host, cross-reactivity would be important to prevent reinfection. In this regard, measuring the cross-reactivity after infection using the ‘neutralization breadth index’ (NBI) has been proposed [20].

This study retrospectively examined (1) the time to SARS-CoV-2 polymerase chain reaction (PCR) conversion and (2) longitudinal neutralizing Ab titers including NPI and NBI up to 8 months after infection in moderate COVID-19 patients treated with or without favipiravir. The data used in this report derived in part from a study conducted by Moriyama et al. [20] supplemented by additional studies conducted to enhance our understanding of the effect of favipiravir on anti-SARS-CoV-2 Ab levels.

## 2. Materials and Methods

### 2.1. Study Design

Patients enrolled in this study were recovered from moderate COVID-19 infection diagnosed at Tokyo Shinagawa Hospital by (1) positive SARS-CoV-2 real-time reverse-transcription polymerase chain reaction (RT-PCR) test using nasopharyngeal swab specimens and (2) positive chest X-ray/computed tomography findings of pneumonia, and visited outpatient department after hospital discharge between 28 May 2020, and 26 September 2020. Disease severity was defined as mild, moderate (I/II), or severe by the Japanese COVID-19 clinical practice guideline ver. 2 [25]. In these criteria, patients having COVID-19 pneumonia with or without respiratory failure (SpO2 ≤ 93%) were classified as moderate II or I, respectively. The patients requiring admission to Intensive Care Unit or mechanical ventilator were classified as severe. To determine SARS-CoV-2 elimination from the host, RT-PCR tests were serially performed every 3 days until hospital discharge or ≥48 h after symptom resolution and/or days between 28–35 after hospital discharge.

### 2.2. Case and Control Definition

A case was defined as patients initiated with compassionate use of favipiravir under informed consent, while controls refused to treat with favipiravir or excluded due to poor performance status, hyperuricemia, or need for dialysis. In these cases, favipiravir was introduced at 3600 mg for the first day, followed by 1600 mg per day for up to 14 days [26]. As reported, characteristics of patients suffering from severe COVID-19 were elderly males with a higher body mass index [27,28], age-, gender-, and body mass index-matched patients treated without favipiravir defined as the control were selected by the MedCalc case-control matching system (MedCalc Version 19). More precisely, controls were individually matched with each case for age (±6 years), gender, and body mass index (±2). In addition, the medical records of each patient were reviewed. The patients with expansion of pulmonary infiltration or requiring oxygen supplementation were treated with methylprednisolone [29,30].

### 2.3. Sample Collection

Enrolled patients were followed up to 13 March 2021. Total 273 blood samples from 99 patients were further processed to obtain plasma and mononuclear cells by using Vacutainer mononuclear cell preparation tubes (BD Biosciences) after centrifugation at 1800× *g* for 20 min, followed by an additional 800× *g* for 15 min.

### 2.4. Neutralizing Antibody Titer Measurement

The virus neutralization was assessed using authentic virus as described previously [20]. Briefly, viral suspension (SARS-CoV-2 JPN/TY/WK-521 or JPN/TY8-612 strain) and Vero E6/TMPRSS2 cells (JCRB1819 VeroE6/TMPRSS2, JCRB Cell Bank) were prepared. First, sera from patients were serially diluted and mixed with viral suspension (100 Tissue Culture Infective Dose (TCID 50)) for 1 h at 37 °C. Finally, the mixed suspension was added onto 1 × 10^4^ of Vero E6/TMPRSS2 cells seeded in 96-well plates and incubated for 1 h at 37 °C followed by further cultured for 4–6 days at 37 °C. Finally, the cultured cells were fixed and stained with crystal violet solution for determining the neutralizing (NT) antibody titer by average from 4–6 wells of cut-off dilution index with >50% cytopathic effect for each sample [20]. Above mentioned experiments using SARS-CoV-2 were performed in a biosafety level 3 facility in National Institute of Infectious Diseases (Tokyo, Japan).

### 2.5. ELISA

RBD IgG-specific enzyme-linked immunosorbent assay system was used to quantify the RBD IgG levels in plasma samples [20]. Briefly, 96-well Nunc-Immuno Plate F96 Maxisorp plates (Thermo Fisher Scientific, Waltham, MA, USA) were coated with 2 μg/mL of recombinant RBD (amino acids: 331–529) overnight at 4 °C and then blocked with PBS/1% BSA. Heat-inactivated plasma and monoclonal antibodies of either COVA1-18 or CR3022 were added with serial dilution and incubated overnight at 4 °C. On the following days, HRP-conjugated Anti-human IgG (Southern Biotech, Birmingham, AL, USA) in Can Get Signal #2 (TOYOBO) was added and then HPR activity was visualized/detected by OPD substrate (Sigma, Kawasaki, Japan). RBD IgG titer in plasma was determined by reference antibody.

### 2.6. Quantification and Statistical Analysis

RBD IgG and NT were log-transformed before all statistical procedures. The NPI was calculated by dividing NT with RBD IgG. The NBI was calculated by dividing the variant (JPN/TY8-612 strain) NT by the WT (JPN/TY/WK-521 strain) NT. According to the data distribution assessed by the Shapiro–Wilk test, continuous variables were expressed as mean (standard deviation; SD) or median (interquartile range; IQR). Paired comparisons were performed by *t*-test or Wilcoxon test as appropriate. Comparisons between categorical variables were analyzed with the chi-square test or Fisher’s exact test. Time-to-event data were calculated by the Kaplan–Meier method, and the log-rank test was used to compare differences between groups. A two-tailed *p*-value of <0.05 was considered statistically significant. The statistical analyses were performed using MedCalc version 19 (Ostend, Belgium).

## 3. Results

### 3.1. Patients Characteristics

Ninety-nine hospitalized patients (mean age 49.5 years ± 16.4 SD) diagnosed with moderate COVID-19 were initially included in a longitudinal study of acquired humoral immunity against SARS-CoV-2. The cohort included 34 patients (mean age 52.9 years ± 14.2 SD) who received the antiviral therapy favipiravir during hospitalization. Of these 34 patients, 17 could be matched (1:1) with controls (from the remaining 65 patients without treatment of favipiravir) based on age (±6 years), gender, and body mass index (BMI) (±2) resulting in the case-controlled study of 34 patients. Their clinical characteristics are summarized in Table 1.

Subjects were followed as outpatients with a first visit 25–150 days after discharge (median 56 (34 to 92) days) and seen for up to 254 days (median 187 (183 to 208) days) after symptom onset. The cohort was biased toward males (76.5%) with ages ranging from 22 to 73 years. The mean (SD) age was 49.6 (14.8) and 50.1 (14.5) for case and control, respectively. Smoking habits and comorbidities such as hypertension and diabetes mellitus are factors known to be associated with COVID-19 severity [31], and these factors were present at similar frequency in each group. Of the cohort, seven patients (five from case and two from control) required supplemental oxygen (the severity classified as moderate II) at the time of admission or during hospitalization. Thus, although the case group included more relatively severe patients, the difference between groups was not statistically significant (*p* = 0.396). The mean levels of oxygen saturation upon admission were 97.1% for case and 96.4% for control, and this was also not statistically different (Table 1, *p* = 0.257). The time to diagnosis from symptom onset was also comparable between groups (Table 1, *p* = 0.186). As for case group, favipiravir was initiated a median of 9 days (range 6–12 days) after symptom onset (Table 1). 

### 3.2. Effect of Favipiravir Treatment on Clinical Outcome

We first analyzed the effect of favipiravir on the (1) time to viral clearance and (2) duration of hospitalization. The use of corticosteroids for hospitalized COVID-19 patients can reduce mortality and decrease ventilator dependence [30,32,33], thus patients received methylprednisolone as needed. The total amount of methylprednisolone used by case and control groups was similar (Table 2, case vs. control, median [IQR] 400 [150 to 1070] vs. 400 [90 to 660] mg, *p* = 0.835). The median dose of methylprednisolone was equivalent to that previously shown to have no impact on viral clearance [34]. Conversion to PCR negatively from the time of symptom onset in the case group was significantly shorter than in the control (median 14.0 (13.0 to 18.0) vs. 18.0 (13.8 to 28.0), *p* = 0.049). A similar effect was seen when focusing on the duration of PCR positively in the case vs. control groups (median 10.0 (6.0 to 12.8) vs. 15.0 (10.5 to 21.3) days, *p* = 0.027). Further, as treatment with favipiravir was initiated on average 1 day after hospitalization, the time to PCR conversion from the time of hospitalization was examined as this measure could show the real power of favipiravir to eliminate virus. As expected, a significant shorter time from hospitalization to PCR negativity was found in the case vs. control groups (median 8.0 [(.0 to 12.0) vs. 11.0 (8.8 to 18.5) days, *p* = 0.039) (Figure 1 and Table 2). However, this did not affect the duration of hospitalization between groups measured from the time of symptom onset to hospital discharge (case vs. control: median 17.0 (15.8 to 21.3) vs. 18.0 (14.0 to 21.5) days, *p* = 0.855) (Table 2).

### 3.3. Effect of Favipiravir Treatment on Long-Term Neutralizing Antibodies Production

A total of 83 sera from 34 patients (median 3 collections/patient) were assessed to quantify neutralizing IgG Ab levels specific for RBD (Figure 2a), NT (Figure 2b), and calculated NPI (Figure 2c) over time. The locally weighted scatterplot smoothing (LOESS) algorithm trends of RBD IgG and neutralizing antibody titers in both case and control groups showed similar time-dependent decreases with an inflection point from rapid to slow decay between 100–200 days after symptom onset (Figure 2a,b). In contrast, the LOESS trend of calculated NPI in both groups rose with an inflexion point in the treated group between 100–200 days from symptom onset, consistent with improved maturation of the anti-SARS-CoV-2 response (Figure 2c). Based on recent reports demonstrating that (1) RBD IgG decayed through week 12 and plateaued at approximately 24 weeks after symptom onset in moderate COVID-19 [35], and (2) NT decreased by half from peak values through 6 months [36], we split our analysis to compare early (≤180 days) versus late (>180 days) phase convalescence by assessing the highest neutralizing antibodies in each phase.

Consistent with previous reports [35,36], RBD IgG levels were significantly higher in early vs. late convalescent phases both in case (median 3.42 vs. 0.86 μg/mL, *p* = 0.0003) and control groups (median 4.52 vs. 0.90 μg/mL, *p* = 0.003). There was no significant different in RBD IgG levels between case and control groups at either time point (early vs. late, *p* = 0.438 vs. *p* = 0.832, Figure 3a). Next, we evaluated NT levels and found they rapidly decay in the control group (early vs. late, median 20 (10 to 20) vs. 5 (3.1 to 17.5), *p* = 0.0002, Figure 3b). In contrast, no significant decay was observed from early (median 10 (5 to 20)) to late (median 10 (5 to 10)) convalescent phases among patients treated with favipiravir (*p* = 0.173). As indicated in Figure 2c, the time-dependent increase in NPI was observed in both case (early vs. late, mean 3.31 vs. 9.26, *p* < 0.0001) and control groups (early vs. late, mean 3.37 vs. 9.74, *p* = 0.031), although the NPI at each phase was comparable between groups (Figure 3c).

### 3.4. Effect of Favipiravir Treatment on Cross-Reactive Humoral Immunity against SARS-CoV-2 Variants

The sera collected from 34 patients were further assessed to quantify antibodies against SARS-CoV-2 variant of B.1.351 (beta) carrying triple mutations of K417N, E484K, and N501Y). The IgG levels against mutant RBD were comparable between the case and control groups in both early and late convalescent phases (Figure 4a). Similar to the response against WT RBD, a time-dependent decrease in anti-mutant RBD IgG was observed in both case and control groups (early vs. late, *p* < 0.0001). NT against the variants was lower than against WT virus especially in early phase regardless of using favipiravir, although levels were maintained in the late phase (Figure 4b). It was interesting to note that the calculated mutant NPI (determined by dividing the mutant NT by mutant RBD IgG) and NBI in the case group was significantly higher compared to those in the control group during the early phase (Figure 4c,d). A time-dependent increase in both mutant NPI and NBI was observed in both case and control groups indicating that favipiravir treatment did not interfere with the cross-reactivity against SARS-CoV-2 variants (Figure 4c,d).

## 4. Discussion

We believe this to be the first matched case-control study exploring the effect of favipiravir on the persistence of humoral immunity in moderate COVID-19 patients. In this study, 17 pairs of age-, gender- and BMI-matched subjects were enrolled before the first isolation of SARS-CoV-2 variants in Japan and followed with serial blood collections to monitor RBD IgG, NT and NPI against both WT SARS-CoV-2 and its variant B.1.351 (beta). The clinical characteristics, including smoking status and comorbidities of participants, were equally distributed between case and control (Table 1). 

Favipiravir is an oral selective inhibitor of viral RNA-dependent RNA polymerase that is approved to treat new or re-emerging influenza virus infection in Japan. Since the drug is safe and targets an element of the viral replication process used by SARS-CoV-2, a phase 3 clinical trial exploring the efficacy of favipiravir in COVID-19 patients with non-severe pneumonia was conducted. Compassionate use of the drug in patients with moderate–severe COVID-19 is also allowed. These clinical efforts showed that favipiravir shortened the time to SARS-CoV-2 PCR negativity (favipiravir vs. placebo: median 11.9 vs. 14.7 days, *p* = 0.0136) [6]. 

The shorter time until viral clearance, noted in other clinical trials [3,37], was reproduced in the current study of favipiravir. Notably, the favipiravir treated cohort showed significant shorter (i) time from symptom onset to PCR conversion (*p* = 0.049), (ii) time from diagnosis to PCR conversion (*p* = 0.027) and (iii) time from treatment initiation to PCR conversion (*p* = 0.039) when compared to the control group (Figure 1 and Table 2). Unfortunately, the favipiravir effect did not shorten hospitalization time probably due to the delay in initiating treatment (Table 2).

The current study examined the kinetics and duration of host humoral immunity in patients with moderate COVID-19 treated with favipiravir. Building on recent evidence involving neutralizing antibodies detected in Japanese patients recovering from COVID-19 who showed sustained protective levels of neutralizing titers 6 months after diagnosis, we divided our subjects into two convalescent phases: before and after 6 months from symptom onset [38]. The levels of RBD IgG during the early convalescent phase (≤180 days) were similar between groups (case vs. control: median 2.59 vs. 4.64 μg/mL, *p* = 0.221) as was their rapid decay from early to late (>180 days) phase (Figure 2a and Figure 3a). In contrast, while NT rapidly fell in controls (20 down to 5, *p* = 0.025), no significant decay was observed in the favipiravir treated group (early vs. late: median 10 (5 to 20) vs. 10 (5 to 10), *p* = 0.173, Figure 2b and Figure 3b). This resulted in the NPI trend lines crossing around 150 days after symptom onset (Figure 2c). Although the NPI in the case vs. control groups did not differ during the late convalescent phase (up to 254 days) (Figure 3c), longer follow-up might be needed because NPI could be a key to long-lasting immunity against SARS-CoV-2 and further cross-reactivity against SARS-CoV-2 variants [20].

Next, we examined whether antiviral favipiravir affected the neutralizing respond against SARS-CoV-2 variants. Results showed that favipiravir did not interfere the generation of cross-reactive antibodies at either the early or late convalescent phases (Figure 4a,b). On the contrary, it should be noted that favipiravir might induce rapid maturation of neutralizing potency against SARS-CoV-2 variants, though its detailed mechanism remains unknown (Figure 4c,d).

Taken together, these results indicate that treatment of COVID-19 with favipiravir accelerates viral clearance and does not interfere the generation or maturation of neutralizing potency against both WT SARS-CoV-2 and its variant B.1.351 (beta). These findings support the conclusion that favipiravir does not increase the risk of early reinfection.

This study has several limitations. First, the trial included more males than females such that some gender-specific differences in humoral responses might have been observed. Second, the case included more severe patients requiring oxygen supplementation, leading limited effect on length of hospital stay. Third, samples were not collected from each member of the cohort at uniform intervals after onset. Fourth, this was a retrospective/observational study involving a small number of patients at a single center. Thus, the generalizability of the current findings should be interpreted with caution. Despite these limitations, given the statistically favorable outcomes observed in patients treated with favipiravir, our results support further study evaluating whether this antiviral therapy positively affect host immunity.

## Figures and Tables

**Figure 1 viruses-14-00670-f001:**
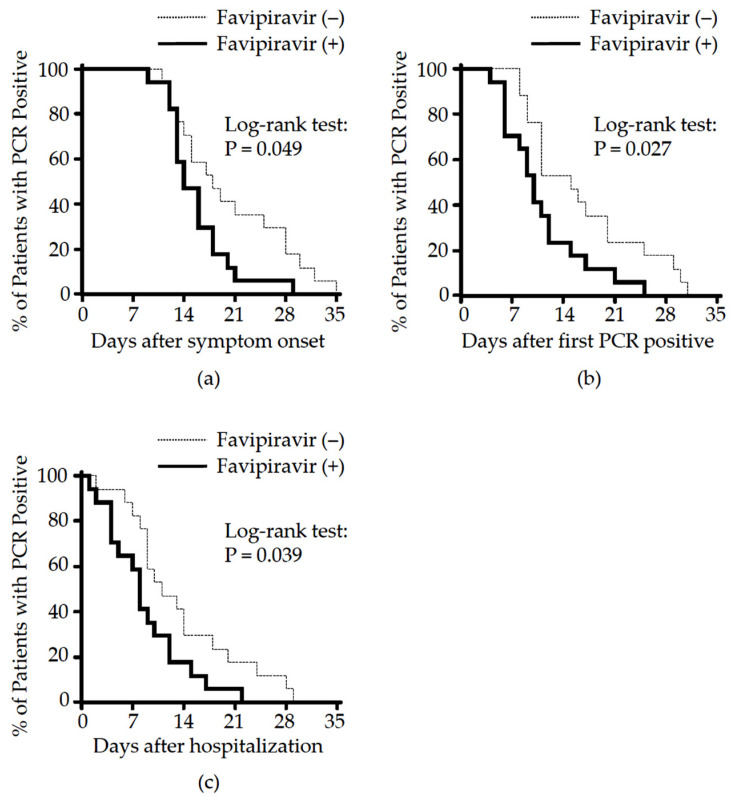
Effect of favipiravir on viral clearance in patients with moderate COVID-19. Kaplan–Meier curves of time to viral clearance from (**a**) symptom onset, (**b**) first day of positive PCR test (equivalent to the terms of PCR positive duration), and (**c**) hospitalization were made for case (solid line) and control (dotted line) and analyzed using the log-rank test.

**Figure 2 viruses-14-00670-f002:**
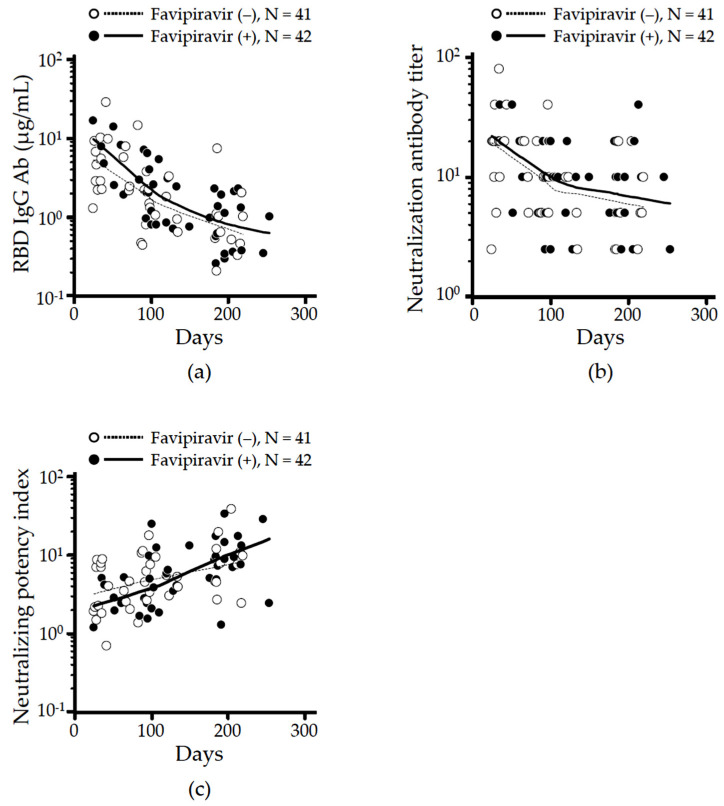
Kinetic analysis of virus-neutralizing antibody production in patients with moderate COVID-19. Longitudinal changes in (**a**) RBD IgG antibody titers, (**b**) neutralizing antibody titers and (**c**) neutralizing potency index in favipiravir treated patients (●) versus controls (◦). Trends were calculated using a locally weighted scatterplot smoothing (LOESS) algorithm and are shown as solid (case) or dotted (control) lines.

**Figure 3 viruses-14-00670-f003:**
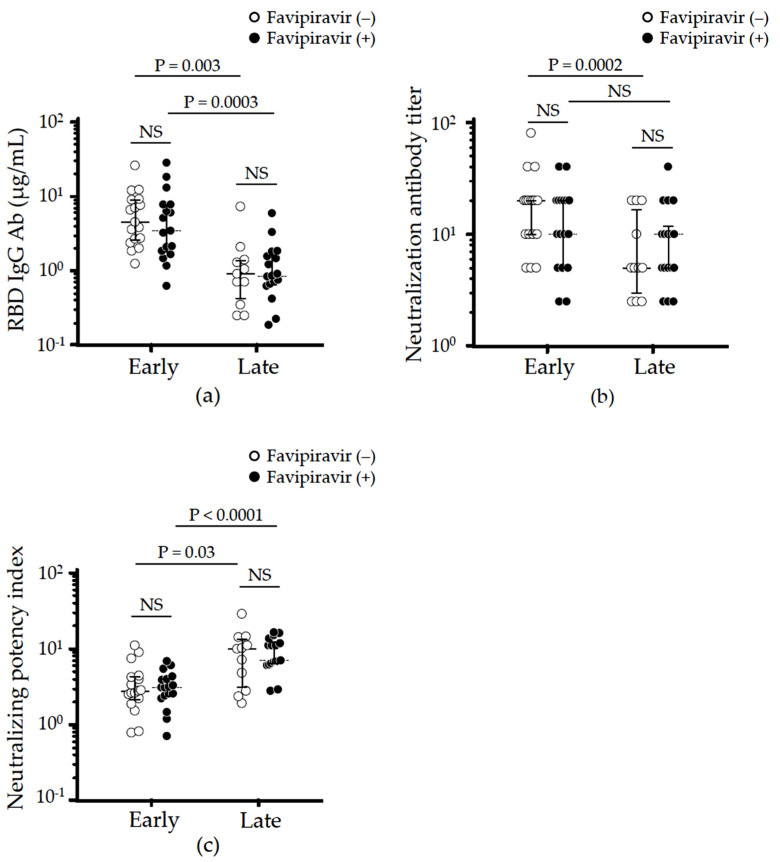
Effect of favipiravir on virus-neutralizing antibody production, duration, and maturation in patients with moderate COVID-19. The highest values of (**a**) RBD IgG antibody production, (**b**) neutralizing antibody titer, and (**c**) neutralizing potency index in early (≤180 days after symptom onset) versus late (>180 days after symptom onset) convalescent serum from each recovered patient were analyzed and compared. Statistical analyses were done using the Wilcoxon test or paired *t*-test as appropriate for early vs. late in the same treatment group and Mann–Whitney U-test or *t*-test as appropriate for case (●) vs. control (◦) in the same time phase. NS: not significant.

**Figure 4 viruses-14-00670-f004:**
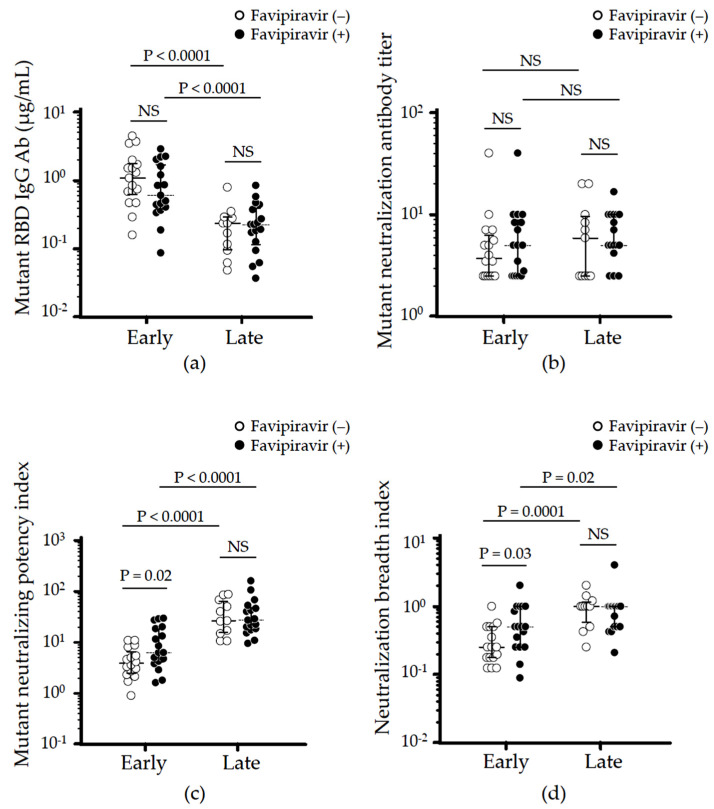
Effect of favipiravir on cross-reactive humoral immunity against SARS-CoV-2 variants carrying K417N, E484K, and N501Y in moderate COVID-19 patients infected with WT SARS-CoV-2. The highest values of (**a**) mutant-RBD IgG antibody production, (**b**) mutant-neutralizing antibody titer, (**c**) mutant-neutralizing potency index, and (**d**) neutralizing breadth index indicating the cross-neutralizing capacity in early (<180 days after symptom onset) versus late (>180 days after symptom onset) convalescent serum from each recovered patient were analyzed and compared. Statistical analyses were done using the Wilcoxon test or paired *t*-test as appropriate for early vs. late in the same treatment group and Mann–Whitney U-test or *t*-test as appropriate for case (●) vs. control (◦) in the same time phase. NS: not significant.

**Table 1 viruses-14-00670-t001:** Baseline characteristics on the admission of included cases and matched controls diagnosed as moderate COVID-19.

Characteristics	All Included Patients (n = 34)	Case (n = 17)	Control (n = 17)	*p* Value *
Age, mean (SD), yrs	49.8 (14.4)	49.6 (14.8)	50.1 (14.5)	0.926 ^a^
Gender, male/female, number (%)	26 (76.5)/8 (23.5)	13 (76.5)/4 (23.5)	13 (76.5)/4 (23.5)	0.686 ^b^
Body mass index, mean (SD), kg/m^2^	23.8 (3.1)	23.9 (3.2)	23.7 (3.1)	0.897 ^a^
Smoking status: current and former/never, number (%)	10 (29.4)/24 (70.6)	5 (29.4)/12 (70.6)	5 (29.4)/12 (70.6)	0.707 ^b^
Comorbidities, any of the listed conditions, number (%)	10 (29.4)	5 (29.4)	5 (29.4)	0.707 ^b^
Hypertension, number (%)	7 (20.6)	4 (23.5)	3 (17.6)	1.000 ^c^
Diabetes, number (%)	4 (11.7)	3 (17.6)	1 (5.9)	0.601 ^c^
Hyperlipidemia, number (%)	3 (8.8)	2 (11.8)	1 (5.9)	1.000 ^c^
Hyperuricemia, number (%)	2 (5.9)	1 (5.9)	1 (5.9)	1.000 ^c^
Hyperthyroidism, number (%)	1 (2.9)	0 (0)	1 (5.9)	1.000 ^c^
Disease severity, moderate I/II, number (%)	27 (79.4)/7 (20.6)	12 (70.6)/5 (29.4)	15 (88.2)/2 (11.8)	0.398 ^c^
Admission oxygen saturation, SpO_2_, median (IQR), %	97.0 (96.0, 98.0)	97.0 (96.8, 98.0)	97.0 (95.0, 98.0)	0.479 ^d^
The time from symptom onset to diagnosis, mean (SD), days	4.2 (2.3)	4.7 (2.3)	3.8 (2.2)	0.238 ^a^
The time from symptom onset to initiate favipiravir, mean (SD), days		8.9 (2.0)	NA	

Abbreviations: SpO_2_, percutaneous oxygen saturation; IQR, interquartile range; NA, not applicable. * Comparison of cases and controls; ^a^ unpaired *t*-test; ^b^ Yates’s chi-squared test; ^c^ Fisher’s exact test; ^d^ Mann-Whitney U test.

**Table 2 viruses-14-00670-t002:** Clinical outcomes of included cases and matched controls diagnosed as moderate COVID-19.

Outcome	All Included Patients (n = 34)	Case (n = 17)	Control (n = 17)	*p* Value *
Total amount of methylprednisolone used, median (IQR), mg	400.0 (120.0, 800.0)	400.0 (150.0, 1070.0)	400.0 (90.0, 660.0)	0.835 ^a^
The time from symptom onset to hospital discharge, median (IQR), days	17.5 (15.0, 21.0)	17.0 (15.8, 21.3)	18.0 (14.0, 21.5)	0.855 ^b^
Hazard ratio (95% CI)		1.070 (0.519, 2.208)		
The time for hospitalization, median (IQR), days	10.5 (8.0, 14.0)	10.0 (7.8, 14.8)	11.0 (8.3, 14.3)	0.831 ^b^
Hazard ratio (95% CI)		0.925 (0.452, 1.893)		
The time from symptom onset to PCR conversion, median (IQR), days	16.0 (13.0, 21.0)	14.0 (13.0, 18.0)	18.0 (13.8, 28.0)	0.049 ^b^
Hazard ratio (95% CI)		0.457 (0.209, 0.998)		
The time from hospitalization to PCR conversion, median (IQR), days	9.0 (7.0, 14.0)	8.0 (4.0, 12.0)	11.0 (8.8, 18.5)	0.039 ^b^
Hazard ratio (95% CI)		0.448 (0.209, 0.960)		
The time for PCR positive duration, median (IQR), days	11.0 (9.0, 17.0)	10.0 (6.0, 12.8)	15.0 (10.5, 21.3)	0.027 ^b^
Hazard ratio (95% CI)		0.412 (0.188, 0.902)		

Abbreviations: IQR, interquartile range; CI, confidence interval. * Comparison of cases and controls; ^a^ Mann–Whitney U test; ^b^ Log-rank test.

## Data Availability

The data presented in the current study are available from the corresponding author upon reasonable request.

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
