# Peer review of "Longitudinal Analysis of Neutralizing Potency against SARS-CoV-2 in the Recovered Patients after Treatment with or without Favipiravir"

_viruses, 2022, doi:10.3390/v14040670_

Round 1

Reviewer 1 Report

The authors reported the analysis of neutralizing antibodies against SARS-CoV-2 in patients treated with faviripavir and in control cohort. Results are very interesting, suggesting that the production and duration of neutralizing antibodies are not influenced by the therapy, also against SARS-CoV-2 variants. The aim of the present study is relevant and the manuscript is clearly presented and written. However some details are missing in the Materials and methods section:

  1. How much virus was used in neutralization assay?
  2. What is the final readout of this assay?
  3. The number of cells used is very high for a 96 well plate. Please revise this information. 

Author Response

Thank you very much for your valuable comments and suggestions.

According to your suggestions, we added more information and now believe that the revised manuscript would be informative to the readers.

  1. How much virus was used in neutralization assay?

R1. We added the information as follows (in revised L.169-170 in “Track Changes” text):

First, sera from patients were serially diluted and mixed with viral suspension (100 Tissue Culture Infective Dose [TCID 50]) for 1 hour at 37℃.

  1. What is the final readout of this assay?

R2. We performed this to be determined the neutralizing (NT) antibody titer for each sample. So, We added more information regarding the determination of NT antibody titer as follows (revised L. 173-175):

Finally, the cultured cells were fixed and stained with crystal violet solution for determining the neutralizing (NT) antibody titer by average from 4-6 wells of cut-off dilution index with >50% cytopathic effect for each sample (Moriyama et al. Immunity 2021).

  1. The number of cells used is very high for a 96 well plate. Please revise this information. 

R3. Thank you again for your comments.

We corrected the number of cells as 1x104 of VeroE6/TMPRSS2 cells in the revised manuscript (L. 171).

Reviewer 2 Report

The manuscript submitted by Shinada et al. reports that the Faviporavir, an oral selective inhibitor of viral RNA-dependent RNA polymerase, treatment significantly decreased the duration of COVID-19. Nevertheless, neutralizing antibodies titers during convalescence against SARS-CoV-2 were not affected with Faviporavir treatment.

Although these findings are interesting, the manuscript contains several issues that need to be addressed.

Major comments

  1. The introduction is, unfortunately, not specific enough and does not clearly give a rationale for the study. I would recommend moving some parts of the discussion (such as “Antiviral therapy… [7,8,32].”, lines 291-295) into the introduction. Moreover, the authors might wish to clearly state what is the current status in the field on Faviporavir use – especially mention the results of the current meta-analyses (i.e. Hassanipour et al. Scientific Reports 2021 11: 11022). I think that it would strengthen the manuscript if the authors had mentioned that Faviporavir use shortens disease duration and/or decreases disease severity (as they show also themselves) and that it is known the amount of neutralizing antibodies is related to the disease severity and/or duration (i.e. Bosnjak et al. Cell Mol Immunol 2021 18:936, Wu et al. Nat Comm 2021 12:1813, etc.) as the reasons for conduction of this study.
  2. The results are clearly presented. Nevertheless, I would suggest moving current subchapter 3.4. “Effect of favipiravir treatment on clinical outcome immediately” after current subchapter 3.1 and before current subchapter 3.2. In this order, first, the effect of the treatment during active disease would be presented and then, after showing that there is a difference between the groups, the results of the antibody titers during convalescence would be presented.
  3. The reported neutralizing titer values seem extremely low when compared to other published studies, even from the same group (Moriyama et al. Immunity 2021). How do the authors explain these discrepancies?

Minor comments

  1. Figure 1 – please include also legend for the lines, as it is not clear to which group of samples which line belongs.
  2. Lines 62-64: Please update the list of SARS-CoV2 variants by mentioning omicron.
  3. Lines 71-73: It seems that the reference is missing. 

Author Response

Thank you very much for your comments and suggestions.

We agree with your suggestions, and our reply to your comments is as follows:

Major comments

1.  The introduction is, unfortunately, not specific enough and does not clearly give a rationale for the study. I would recommend moving some parts of the discussion (such as “Antiviral therapy… [7,8,32].”, lines 291-295) into the introduction. Moreover, the authors might wish to clearly state what is the current status in the field on Faviporavir use – especially mention the results of the current meta-analyses (i.e. Hassanipour et al. Scientific Reports 2021 11: 11022). I think that it would strengthen the manuscript if the authors had mentioned that Faviporavir use shortens disease duration and/or decreases disease severity (as they show also themselves) and that it is known the amount of neutralizing antibodies is related to the disease severity and/or duration (i.e. Bosnjak et al. Cell Mol Immunol 2021 18:936, Wu et al. Nat Comm 2021 12:1813, etc.) as the reasons for conduction of this study.

R1. As recommended, the revised introduction now includes

1) The concerns of antiviral therapy could affect the host immune response as “antiviral therapy could affect the host immune response by decreasing the amount and duration of viral antigen, potentially influencing subsequent susceptibility to reinfection. For example, treatment with anti-influenza virus drugs reduced production of mucosal secretory IgA and protective Abs at both early (21 days) and late (60 days) times after influenza infection in murine models, that may account for the higher reinfection rates observed in patients treated with oseltamivir or zanamivir vs untreated controls the following year [11-13].” (in revised “Track Changes” L.55-61)

2) The current status of the antiviral Favipiravir usage for COVID-19 as “As of Mar 17, 2022, a total of 24 phase 3 clinical studies exploring the effect of favipiravir on COVID-19 in over 20 countries were registered at ClinicalTrials.gov [7]. A recent meta-analysis of 9 favipiravir clinical trials showed significant clinical improvement within 7 days of hospitalization in the favipiravir group (p = 0.001 vs. control group) [8]. As for the antiviral effects of favipiravir, faster viral clearance was observed although the difference did not reach statistical significance (p = 0.094) in this meta-analysis [8]. A more sophisticated phase 3 clinical trial with a larger sample size involving early-onset COVID-19 patients with risk factors has been initiated to examine these issues in greater detail [9,10].” (in revised “Track Changes” L.38-51)

3) The revised introduction includes more information on the key neutralization factor of “neutralizing Abs” that we focused on in this study as “Key to the neutralization of SARS-CoV-2 are Abs specific for the receptor binding domain (RBD) by blocking cell entry of SARS-CoV-2, while the lower levels of anti-RBD Abs associated with mild disease and/or shorter duration of symptoms [17,18]. Therefore, concerns about the magnitude of host immune activity against COVID-19 after favipiravir therapy should be addressed.” (in revised “Track Changes” L.68-72)

2.  The results are clearly presented. Nevertheless, I would suggest moving current subchapter 3.4. “Effect of favipiravir treatment on clinical outcome immediately” after current subchapter 3.1 and before current subchapter 3.2. In this order, first, the effect of the treatment during active disease would be presented and then, after showing that there is a difference between the groups, the results of the antibody titers during convalescence would be presented.

R2. We appreciate your thoughtful suggestion and agree with your proposed modification of the presentation order. Thus, we would like to inform you that the subchapter “3.4. Effect of favipiravir treatment on clinical outcome” appears just after subchapter 3.1. and Figure 4 is now renumbered as Figure 1. Accordingly, the original Figures 1 to 3 are also renumbered as Figures 2 to 4 in revision. As you recommended, the readers could primarily understand the effect of the antiviral favipiravir and further recognize the result does not interfere with humoral antiviral immunity.

3.  The reported neutralizing titer values seem extremely low when compared to other published studies, even from the same group (Moriyama et al. Immunity 2021). How do the authors explain these discrepancies?

R3. We used the live virus of JPN/TY/WK-521 strain (WT) and JPN/TY8-612 strain (mutant) for determining NT titers. As for JPN/TY/WK-521 strain, the NT titers in early phase were median of 20 (95% CI 10, 20) for control and of 10 (95% CI 5, 20) for favipiravir group (original Figure 2b). As you mentioned, Moriyama et al. showed higher NT titers (mainly distributed between 100 and 1000) in the assay using SARS-CoV-2 pseudovirus (Figure 1D in Immunity 2021;54:1841-1852), whereas those using live virus of JPN/TY/WK-521 strain were mainly distributed between 5 to 40 in patients with moderate COVID-19 (Figure 4A and S3A/B in Immunity 2021;54:1841-1852). Thus, we think that the NT values of our current and previous reports using authentic virus assay are comparable.

The differences of the NT values between pseudovirus- and live virus-assay could be due to the differences of endpoint measurements (IC50 for pseudovirus assay and cytopathic effect of <50% for the live virus), however, the correlation was confirmed in our previous report (Figure S3A in Immunity 2021;54:1841-1852).

Minor comments

1.  Figure 1 – please include also legend for the lines, as it is not clear to which group of samples which line belongs.

R1. Thank you for the comment. We added in revised Figure 2.

2. Lines 62-64: Please update the list of SARS-CoV2 variants by mentioning omicron.

R2. We added relevant references in revision.

3. Lines 71-73: It seems that the reference is missing. 

R3. Thank you for pointing it out. We added the reference (Moriyama et al. Immunity 2021, 54, 1841-1852).